# Robust Independent Validation of Experiment and Theory: RIVET version 4 release note

C. Bierlich[1], A. Buckley[2], J. Butterworth[3], C. Gütschow[3,4,*], L. Lönnblad[1],
T. Procter[2], P. Richardson[6], Y. Yeh[3],

[1] *Department of Physics, Lund University*
*Professorsgatan 1, 223 63 Lund, Sweden*
[2] *School of Physics & Astronomy, University of Glasgow,*
*University Place, G12 8QQ, Glasgow, UK*
[3] *Department of Physics & Astronomy, University College London,*
*Gower Street, WC1E 6BT, London, UK*
[4] *Centre for Advanced Research Computing, University College London,*
*Gower Street, London, WC1E 6BT, UK*
[5] *Institute for Particle Physics Phenomenology, Durham University*
*South Road, Durham, DH1 3LE, UK*

May 14, 2024

## Abstract

The RIVET toolkit is the primary mechanism for phenomenological preservation of collider-physics measurements, containing both a computational core and API for analysis implementation, and a large collection of more than a thousand preserved analyses. In this note we summarise the main changes in the new RIVET 4 major release series. These include a major generalisation and more semantically coherent model for histograms and related data objects, a thorough clean-up of inelegant and legacy observable-computation tools, and new systems for extended analysis-data, incorporation of preserved machine-learning models, and serialization for high-performance computing applications. Where these changes introduce backward-incompatible interface changes, existing analyses have been updated and indications are given on how to update new analysis routines and workflows.

**Analysis-routine authors since RIVET 3**

M. I. Abdulhamid, M. Alvarez, S. H. An, M. Azarkin, L. I. E. Banos, S. Bansal, F. Barreiro, A. Basan, E. Berti, B. Bilin, A. Borkar, V. R. Bouza, I. Bubanja, T. Burgess, K. Butanov, S. Calvente, F. Canelli, J. J. J. Castella, D. S. Cerci, S. Cerci, A. J. Chadwick, M. Chithirasreemadam, J. E. Choi, S. Choi, P. L. S. Connor, K. Cormier, L. D. Corpe, F. Curcio, M. Davydov, A. E. Dumitriu, A. Egorov, S. Epari, F. Fabbri, S. Farry, J. P. Fernandez, K. M. Figueroa, E. Fragiacomo, A. B. Galvan, V. Gavrilov, M. Giacalone, A. Gilbert, D. Gillberg, M. Goblirsch, J. Goh, A. R. C. Gomez, G. Gomez-Ceballos, P. Gras, A. Grebenyuk, A. Grecu, P. Gunnellini, D. Gunther, R. Gupta, R. Gupta, H. V. Haevermaet, J. Haller, R. Hawkings, I. Helenius, O. Hindrichs, A. Hinzmann, D. Hirschbuhl, I. Hos, A. Hrynevich, J. Jahan, J. Jamieson, D. Jeon, C. Johnson, L. S. Johnson, K. Joshi, H. Jung, I. Kalaitzidou, D. Kalinkin, D. Kar, E. Kasimi, L. Keszeghova, H. Kim, S. Kim, T. J. Kim, V. Kim, V. Kireyeu, F. Kling, A. H. Knue, O. Kodolova, K. Koennonkok, R. Kogler, P. Kokkas, L. Kolk, M. Komm, J. Kretzschmar, V. Lang, A. Laurier, J. Lawhorn, J. S. H. Lee, K. Lee, S. Lee, A. Leon, D. Lewis, Y. Li, J. v. d. Linden, K. Lipka, Q. Liu, J. Llorente, K. Lohwasser, K. Long, D. Lopes, Y. Lourens, G. Marchiori, L. Marsili, A. B. Martinez, N. L. Martinez, P. V. Mechelen, Meena, S. Michail, H. Mildner, K. Mishra, O. Miu, U. Molinatti, S. T. Monfared, L. Moureaux, D. Muller, V. Murzin, M. Muskinja, B. Nachman, R. Naranjo, V. Oreshkin, D. Osite, R. Ospanov, P. Ott, M. Owen, S. Pak, S. Palazzo, M. M. d. Melo Paulino, E. Pfeffer, S. Pflitsch, M. Pieters, G. Pivovarov, G. Poddar, G. Pokropska, C. Pollard, K. Rabbertz, M. Radziej, N. Rahimova, S. Rappoccio, J. Robinson, Y. J. Roh, J. Roloff, L. Rossini, F. L. Ruffa, G. Safronov, E. Sauvan, M. Schmelling, M. Schoenherr, D. Schwarz, M. Seidel, S. Sen, F. Sforza, J. Shannon, M. H. K. Sichani, G. Sieber, F. Siegert, R. Sikora, A. Silva, J. B. Singh, S. K. Singh, M. Sirendi, P. Sommer, P. Spradlin, M. Stefaniak, T. Strozniak, C. Sueslue, L. A. Tarasovicova, S. Todt, N. Tran, A. Vaidya, A. Verbytskyi, S. Wahdan, P. Wang, R. Wang, D. Ward, N. Warrack, S. Weber, S. Wertz, S. L. Williams, M. Wing, D. Wislon, M. Wu, D. Yeung, H. Yin, L. Yue, H. Yusupov, W. Zhang, C. Zorbilmez

# 1 Introduction

The RIVET toolkit [1, 2] is the primary mechanism for phenomenological preservation of collider-physics measurement analyses, most prominently but by no means exclusively at the LHC.

It consists of a computational core for theoretically robust reproduction of experimental data-processing, including a configurable system for smearing-based detector emulation, and a large library of more than 1500 preserved analyses. Any event generator supporting the HepMC 3 event standard [3] (either in a serialized data format or in memory) can compare its predictions to hundreds of analyses with minimal interfacing.

RIVET 4 is a new major release series, significantly refining and extending the treatment of data objects, providing mechanisms for efficient parallel running on MPI clusters and for inclusion of machine-learning analysis components, and improving the observable-computation tools. These changes have necessitated some backward-incompatible changes to the programming interface (or "API") exposed to analysis-author users, as well as extensions to the v3 functionality. In this note we summarise the major developments, as well as directing users to what needs to be done to adapt to the API changes.

## 2   Histogramming

RIVET's histogramming system has been reworked to support the new major release series of the YODA 2 [4] statistical data-analysis library. This has significantly simplified RIVET's internal YODA interface, while extending histogramming support to arbitrary histogram and profile dimensions based on modern C++ template techniques. RIVET retains the ability to coherently synchronise correlated NLO subevents into histograms, in addition to handling arbitrary numbers of systematic-variation weight streams; this has similarly been generalised to arbitrary histogram dimensions.

RIVET 4 fully embraces the concept of *live* and *inert* data objects introduced in YODA 2. All of the experimental reference data shipped with RIVET is now represented in terms of `Estimate`-type objects, which lend themselves better to measurement data than the `Scatter`-type objects. This addresses a long-standing issue where intrinsically discrete observables such as multiplicity distributions had to be represented with dummy bin-widths around integer values: while `Scatter`s had to be heuristically interpreted as implying binnings, the `Estimate`s implement continuous or discrete data-binning as appropriate to the data. The new object types are derived from the same underlying `BinnedStorage` formalism championed by YODA 2.

In the same spirit, any *live* types of analysis objects left at the end of the `finalize()` stage will be automatically converted to *inert* estimate types, usually a `BinnedEstimate1D`. This ensures full type consistency and direct comparability between the objects produced in a RIVET run and the reference data. Pre-`finalize` copies of analysis of all analysis data-objects, whose paths are prefixed with `/RAW`, remain live types, allowing re-entrant finalization – i.e. statistically exact merging or extension of MC runs – via the `rivet-merge` command-line utility. Custom conversions to inert form can be explicitly performed in `finalize()`, e.g. by using the `barchart()` function to bypass the default scaling by histogram bin widths.

The confusing `BinnedHistogram` type has been replaced by a new `HistoGroup` class, which better reflects that this object is essentially a binning of histograms. While the new type has the same aim and basic design, it is not implemented as a disjoint standalone class, but rather exploits the fact that YODA 2's new `BinnedStorage` class is agnostic about its templated bin content, and could itself derive from `BinnedStorage`, thereby realising a literal histogram-of-histograms.

As well as the statistical core of the generalised YODA live and inert histogramming, interfaced with RIVET's counter-event and weight-stream coordination, RIVET 4 features a new histogram rendering system. This is built on YODA 2's `matplotlib` plotting system, and replaces the venerable make-plots script based on LaTeX-`pstricks`. The latter produced high-quality output, but was also inflexible and sometimes unstable due to the antiquated backend; the new version reproduces the previous style almost exactly via styling additions to the YODA defaults, within the more modern and supported `matplotlib` ecosystems, and maintains the publication-friendly model of also writing intermediate script files for customisation. Finer details of the cosmetic layout are defined in dedicated `plot` files, so as to not distract from the physics in the main analysis plugins. Existing style files are still supported by the new plotting infrastructure, but newer versions based on the widely used YAML format are now also accepted.

# 3 Projection streamlining

Many of the existing `enum`s have been renamed and often decoupled from specific projection scopes to present a more uniform, predictable, and less deeply nested interface for projection configuration. Projection constructors and methods with unclear `bool` arguments have been converted to use these more "self-documenting" `enum`s. Scoped `enum` classes are now used as standard, enforcing type consistency while improving readability and self-documentation, and are defined outside the projection scopes to reduce the amount of scope chaining.

This simple but widespread change eliminates the previous inconsistent mix of two- and three-layered `enum`s, for instance from both the `JetAlg` and `FastJets` scopes when configuring jet finders. Examples include the relabelling of `FastJets::Algo::KT` to `JetAlg::KT`, of `JetAlg::Muons::NONE` to `JetMuons::NONE`, or from `JetAlg::Invisibles::DECAY` to `JetInvisibles::DECAY`. Similarly, the boolean arguments triggering whether or not tau and muon decay particles are treated as prompt have been replaced by `TauDecaysAs` or `MuDecaysAs enum` classes, respectively.

The `DressedLeptons` projection has been renamed `LeptonFinder`, bringing it more in line with the existing `ParticleFinder` and `JetFinder` classes while being agnostic about the dressing logic used to reconstruction the leptons. Note that the old `DressedLeptons` class name is now an alias for a `vector` of `DressedLepton` objects, which again streamlines the type-naming semantics with respect to the existing `Particles` and `Jets` aliases.

The `ZFinder` has been similarly canonically renamed to `DileptonFinder`, better reflecting both the mass-generality and lepton-specificity of its approach. The constructor arguments have been simplified and the previously implicit target mass is now a explicitly passed constructor argument. The old and often misunderstood `trackPhotons` argument has been removed – the same strategy can be realised by setting the `dR` argument to `-1`.

The `WFinder` has been removed entirely as it had too many analysis options, reflecting the need for analysis-specific heuristics to compensate for the imperfect information induced by the invisible neutrino. It is preferable to implement the analysis-specific selection cuts manually in the analysis, e.g. using dressed leptons and missing tranverse energy explicitly via the new `closestMatchIndex()` metafunction used with the `mass()` or `mT()` unbound functions as appropriate. Applying this migration to the standard analysis collection was found to significantly improve the self-documentation of analysis code, clarifying previously obscure model-dependent assumptions woven into the measurement data.

Finally the jet-smearing system via the `SmearedJets` projection now accepts an ordered list of smearing and efficiency functions via a C++ parameter pack. This has required a re-ordering of parameters in a slightly non-optimal fashion, but was considered an overall preferable solution, especially given that this feature is a somewhat niche requirement for cases where multiple distinct detector effects need to be applied in a specific order.

A FAQ document supplying detailed advice and mechanics for migration of RIVET 3 analysis routines to the v4 API is provided on the RIVET code-development tracker, and linked from the website documentation. As always, any analyses supplied to the team for inclusion in the standard collection are maintained and migrated by the RIVET team, but support is available to all who need help with updating their codes to the new release series.

# 4 Additional interface improvements

Older existing analyses previously named with a SPIRES-based identifier have now been renamed to match newer analyses whose name is already based on their INSPIRE code.

Previous aliasing to allow loading via the INSPIRE name has now been switched such that the non-canonical alias is the SPIRES name. These fallback aliases will eventually be removed.

Building on the similar feature in YODA 2, this release of RIVET introduces a serialization mechanism for the core `AnalysisHandler` objects, so their metadata and list of analysis objects can be efficiently rendered into a stream of `double`-valued numbers, and then reconstructed. While of potentially general usefulness, this mechanism allows synchronisation of many different parallel RIVET instances in an MPI-based high-performance computing cluster, without requiring very expensive access to the filesystem. This mechanism has been demonstrated as an effective component of combined high-precision event generation, showering, and analysis on leading HPC clusters [5–7]. Running with multiple simultaneous `AnalysisHandler` objects across many threads is also now possible, with ability to merge the multiple handlers at the end of the run.

RIVET 4 adds functionality for storing and loading analysis-specific structured data in the widely used `HDF5` format, making `HDF5` as well as the lightweight `HighFive` I/O layer strict dependencies in the process. The (new) canonical analysis name is used as a filename prefix to ensure specificity to the associated analysis routine, and grouping of all files for each analysis in directory listings.

A specialisation of this analysis-data mechanism has been developed to assist import of machine-learning (ML) models, in particular deep neural-net and graph neural-net architectures, from ONNX serialisation files stored specific to the analysis. While some question marks remain over the long-term stability of ML models, e.g. with respect to evolution of their implementing frameworks, ONNX has been identified as the current best option for ML preservation [8]. This option is implemented as a header file only, so the ONNX Runtime [9] library is only an optional dependency for RIVET; if desired, tighter ONNX Runtime integration can be enabled at build-time, so its compiler and linker flags are automatically propagated to the `rivet-build` script, and the currently small collection of ONNX-using analysis routines will be built and installed. Subject to the success of ONNX-preservation initiatives in collider-physics collaborations, this dependency may become mandatory in a future release in the RIVET 4 series.

For simplification of interfacing code, and to reflect its uptake and development status as the new standard particle-level event format in the high-energy physics community, support has been removed for the HepMC 2 API and library structure: the HepMC 2 file format can still be read, via the I/O routines of the HepMC 3 library.

## 5   Conclusions and outlook

The new release series of the RIVET toolkit introduces a number of API-breaking changes, in order to better support more coherent workflows and ensure more self-documenting analysis routines. Additional generalisation provide official mechanisms to load complex auxiliary analysis data, including ONNX machine-learning models, and to use RIVET efficiently in HPC environments. The overall effect of these collective changes is a very positive evolution in RIVET's interface and capabilities, cementing the characteristics that have made it a successful tool for collider-physics analysis preservation. As with all previous versions, RIVET 4 is available for installation from source and as Docker images, as detailed at `https://rivet.hepforge.org/`.

# 6  Acknowledgements

The authors thank the Marie Sklodowska-Curie Innovative Training Network MCnetITN3 (grant agreement no. 722104) for funding and providing the scope for discussion and collaboration toward this work. AB and CG acknowledge funding via the STFC experimental Consolidated Grants programme (grant numbers ST/S000887/1 & ST/W000520/1 and ST/S000666/1) & ST/W00058X/1), and the SWIFT-HEP project (grant numbers ST/V002562/1 and ST/V002627/1). YY thanks the Spreadbury Fund and the UCL Impact scheme for PhD studentship funding. Many thanks to Markus Seidel, Alex Grecu, and Antonin Maire for support as additional LHC experiment contacts. Our thanks also to all the many user-contributors whose inputs and support have enabled RIVET to grow and evolve, and who provide such a welcoming user community, in particular Enrico Bothmann, Christian Holm Christensen, Stefan Höche, Dmitry Kalinkin, Stefan Kiebacher, Max Knobbe, Alexander Puck Neuwirth, Marek Schönherr, Andrii Verbytskyi, and James Whitehead.

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
