# Peer review of "Robust Independent Validation of Experiment and Theory: Rivet version 4 release note"

_SciPost Physics Codebases, doi:SciPost Phys. Codebases 36-r4.0 (2024) , SciPost Phys. Codebases 36 (2024)_

## Round 2 · Referee Report · Anonymous (Referee 1) · 2024-7-5

Strengths

1- The paper explains the updates to the code and the workflow in Rivet in version 4. Sections 2 & 3 explain in detail to those with experience in Rivet, both, the technical changes and the motivation. The explanations are clear and to the point. 2- Section 4 discusses technical advancements, most prominently the inclusion of ONNX support. This option will likely be widely used in HEP.

Weaknesses

1- Section 1 misses a short description of the rivet workflow. While that is discussed in detail in the main publication, a short summary would make the document more readable. 2- Section 4 misses an implementation example. It would be nice to see how ONNX support is used in practice.

Report

The paper fits the journal, but the template must be changed - it is "SciPost Physics" which is confusing.

Requested changes

1- A short description of the rivet workflow, for example in Section 1. 2- A short example of the ONNX feature in Section 4.

Recommendation

Ask for minor revision

  • validity: top
  • significance: top
  • originality: top
  • clarity: top
  • formatting: perfect
  • grammar: perfect

Author:  Christian Gutschow  on 2024-08-14  [id 4694]

(in reply to Report 1 on 2024-07-05)

We thank the referee for the careful reading of the release note.

1- Section 1 misses a short description of the rivet workflow. While that is discussed in detail in the main publication, a short summary would make the document more readable.

--- We've extended the section to include a short summary of the workflow.

2- Section 4 misses an implementation example. It would be nice to see how ONNX support is used in practice.

--- We've added a small example snippet to illustrate the usage.

The paper fits the journal, but the template must be changed - it is "SciPost Physics" which is confusing.

--- We've replaced the style template, thanks for spotting!

Anonymous on 2024-08-27  [id 4719]

(in reply to Christian Gutschow on 2024-08-14 [id 4694])

I found the new draft on the arxiv as v3. I think the submission links the original v2.

Thank you for the updated draft, I am happy with all the changes.

---

## Round 2 · Referee Report · Anonymous (Referee 2) · 2024-8-8

Strengths

  1. A reasonably clear overview of the scope of Rivet 4 is given.
  2. All of the important Rivet 4 changes with respect to Rivet 3 seem to have been covered.
  3. The references seem reasonably complete.

Weaknesses

  1. The paper presupposes familiarity with Rivet 3 and its programming constructs. This is not per se a weakness, but it shuts out newcomers completely. The paper would be helpful to newcomers to Rivet if the introduction were expanded slightly by including pointers to help newcomers get started, in particular, pointers to help newcomers gain conceptual understanding of Rivet before they attempt to dive into code.
  2. The paper points to an FAQ explaining how to migrate from Rivet 3 to Rivet 4, but it would be helpful to have a few sentences explaining to a newcomer who starts with Rivet 4 where she or he should start. It would be helpful to advise whether starting with Ref. [1] is still a good place to begin to learn Rivet or whether given the substantial changes between Rivet 3 and 4 a newcomer is better advised to start elsewhere.

Report

The manuscript is perfectly acceptable for this journal.

Requested changes

The authors should consider the comments under "weaknesses".

Recommendation

Publish (meets expectations and criteria for this Journal)

  • validity: top
  • significance: top
  • originality: high
  • clarity: good
  • formatting: excellent
  • grammar: excellent

Author:  Christian Gutschow  on 2024-08-14  [id 4695]

(in reply to Report 2 on 2024-08-08)

We thank the referee for the careful reading of the release note.

  1. The paper presupposes familiarity with Rivet 3 and its programming constructs. This is not per se a weakness, but it shuts out newcomers completely. The paper would be helpful to newcomers to Rivet if the introduction were expanded slightly by including pointers to help newcomers get started, in particular, pointers to help newcomers gain conceptual understanding of Rivet before they attempt to dive into code."

--- We've expanded the intro somewhat, but prefer not to go into the same level of detail as the original Rivet papers, since this is meant to be a release note rather than a standalone Rivet paper. We have supplied citations for where to get the (still relevant) full introduction from, though.

  1. The paper points to an FAQ explaining how to migrate from Rivet 3 to Rivet 4, but it would be helpful to have a few sentences explaining to a newcomer who starts with Rivet 4 where she or he should start. It would be helpful to advise whether starting with Ref. [1] is still a good place to begin to learn Rivet or whether given the substantial changes between Rivet 3 and 4 a newcomer is better advised to start elsewhere."

--- We've clarified the relevant section to point to the online tutorial which is the best place to get started. Ref. [1] is still a good resource, but serves less as a practical guide.

---

## Round 2 · Referee Report · Anonymous (Referee 1) · 2024-8-27

Report

Thank you for addressing all the comments, I have nothing else to add.
I found the new draft as v3 on the arxiv, but didn't see it linked to Scipost which only seems to link to the initial v2.

Recommendation

Publish (surpasses expectations and criteria for this Journal; among top 10%)

---

## Round 3 · Author Response

We thank the referees for their careful reading of this release note.
We addressed the feedback in this resubmission (v3 on the arXiv).

---

## Round 3 · List of Changes

1. Replaced the SciPost style template
2. Expanded the paper intro
3. Added a short summary of a typical workflow
4. Clarified that the tutorial pages are the best place to start for newcomers
5. Added an example snippet to illustrate ONNX usage

---

## Editorial Decision

published